# DRM-Based Colour Photometric Stereo Using Diffuse-Specular Separation for Non-Lambertian Surfaces

**DOI:** 10.3390/jimaging8020040

**Published:** 2022-02-08

**Authors:** Boren Li, Tomonari Furukawa

**Affiliations:** 1Beijing Institute for General Artificial Intelligence, Beijing 100124, China; liboren@vip.163.com; 2School of Engineering and Applied Science, University of Virginia, Charlottesville, VA 22904, USA

**Keywords:** photometric stereo, dichromatic reflectance model, diffuse-specular separation, non-lambertian surfaces

## Abstract

This paper presents a photometric stereo (PS) method based on the dichromatic reflectance model (DRM) using colour images. The proposed method estimates surface orientations for surfaces with non-Lambertian reflectance using diffuse-specular separation and contains two steps. The first step, referred to as diffuse-specular separation, initialises surface orientations in a specular invariant colour subspace and further separates the diffuse and specular components in the RGB space. In the second step, the surface orientations are refined by first initialising specular parameters via solving a log-linear regression problem owing to the separation and then fitting the DRM using Levenburg-Marquardt algorithm. Since reliable information from diffuse reflection free from specularities is adopted in the initialisations, the proposed method is robust and feasible with less observations. At pixels where dense non-Lambertian reflectances appear, signals from specularities are exploited to refine the surface orientations and the additionally acquired specular parameters are potentially valuable for more applications, such as digital relighting. The effectiveness of the newly proposed surface normal refinement step was evaluated and the accuracy in estimating surface orientations was enhanced around 30% on average by including this step. The proposed method was also proven effective in an experiment using synthetic input images comprised of twenty-four different reflectances of dielectric materals. A comparison with nine other PS methods on five representative datasets further prove the validity of the proposed method.

## 1. Introduction

Photometric stereo (PS) estimates surface orientations using images captured from a fixed viewpoint under various illuminations and is especially powerful in acquisition of fine surface details at pixel level [1]. Surface orientation is important in a variety of fields, such as geometry segmentation for three-dimensional (3D) object recognition [2] and digital re-rendering in computer graphics. Surface geometries, which can be obtained via integrating surface orientations, have also been proven useful for applications, such as industrial quality control and reverse engineering. Due to the strength of PS and the importance of surface orientation acquisition, PS has drawn increasing interests since its debut [3]. However, making PS for a general real scene remains challenging due to the diverse reflectance properties of different materials that appear non-Lambertian [4]. This has given rise to the need for reliable estimation of surface orientations for a wide range of non-Lambertian reflectance, which essentially requires a proper imaging photometry model characterizing the forward problem and a subsequent PS method that inversely derives surface orientations.

Existing PS methods dealing with non-Lambertian reflectance can be classified into three categories. The first approximates surface reflectance using analytical bidirectional reflectance distribution function (BRDF) [5] and formulates the estimation of surface orientations as a nonlinear fitting problem. Nayar et al. [6] derived surface orientations using the Torrance-Sparrow BRDF [7] and incorporated extended sources to ensure sufficient information from specularities. Georghiades [8] inverted the same BRDF to simultaneously estimate surface orientations and resolve the generalized bas-relief ambiguity. Goldman et al. [9] assumed that general material reflectance can be represented by a convex combination of fundamental materials characterized by the Ward BRDF [10] and recovered surface orientations, fundamental material BRDFs and weight maps simultaneously for further scene editing purpose. Methods in this category exploiting information from surface reflectance are capable to derive not only surface orientations but also the other parameters in the analytical BRDFs, allowing more functionalities, such as digital relighting and material classification. However, due to the nonlinearity of analytical BRDFs and larger number of parameters to be estimated, these methods are sensitive to initialisations, numerically unstable under heavily corrupted outliers (e.g., shadows), and inapplicable when the number of observations is limited.

The second category of methods infers surface orientations through adopting the general properties of BRDF, such as isotropy, monotonicity and reciprocity. Alldrin et al. [11] developed a non-parametric PS method using bi-variate approximation of the isotropy property. Higo et al. [12] analyzed the general BRDF constraints and employed the BRDF properties of monotonicity, isotropy and visibility to vote for the most possible surface orientations for single-lobed reflectance. Shi et al. [13] proposed a bi-polynomial representation for low-frequency reflectance that was especially adaptable to the inverse problem as PS, while Ikehata et al. [14] developed another general isotropic BRDF as sum of lobes with unknown center directions. Methods in this category capitalize on the most fundamental properties of BRDF and therefore have the potential to deal with a broader range of reflectance. However, these methods are only capable to derive surface orientations with limited other functionalities and require an even larger set of observations compared with the first category. To widen the applicability, recent years have seen non-parametric BRDFs based on machine learning. Santo et al. [15,16] used a deep neural network for the first time whereas Taniai and Maehara [17] estimated surface normals and BRDFs by unsupervised learning. Ikehata [18] estimated surface normals more straightforward by deriving the so-called observation maps and using convolutional neural networks. Such data-driven approaches clearly improves the accuracy as observations directly create the BRDF or estimate surface normals. The use of a significantly large set of observations, which is key to the accuracy, and the expensive training effort using the observations make the approaches outside the scope of this paper.

Methods in the third category assume that non-Lambertian effects appear sparsely among observations and treat them as outliers. A substantial corpus of methods rely on robust statistical techniques for outlier rejection. Wu et al. [19] formulated the PS with outlier rejection as a global rank minimization problem, while Ikehata et al. [20] employed sparse Bayesian regression instead. Barsky et al. [21] initiated another line of PS researches using colour images where they first identified the significance of using specular colour in the dichromatic reflectance model (DRM) [22] for specularity rejection. Using the cue from known specular colour under the same theoretical foundation, Zickler et al. [23] derived a PS method in a novel two-dimensional (2D) specular invariant colour subspace. The major advantages of methods in the third category are their robustness and requirement for less images, while they are inefficient in the presence of dense non-Lambertian effects.

This paper presents a colour PS method to estimate surface orientations using diffuse-specular separation dealing with non-Lambertian reflectance. The proposed method models the colour imaging photometry using DRM as the forward problem and inverts it following a two-step procedure, the diffuse-specular separation and the surface normal refinement. The first step, initialising surface orientations using known specular colour similarly to [23], separates the diffuse and specular components in the RGB space and identifies outliers to reject in the UV space. In the second step, the parameters characterizing the specularity are initialised via solving a log-linear regression problem and the surface orientations are finally refined by fitting the nonlinear DRM using Levenburg-Marquardt algorithm (LMA). The proposed method robustly initialises surface orientations in a specular-free colour subspace and further fits for the nonlinear DRM with a newly derived parameter initialisation strategy. The proposed method preserves the advantage of robustness as the third category of methods, while it also benefits from the separated specular component to tackle with dense non-Lambertian reflectance as the first category. Furthermore, DRM parameters besides the surface orientations can be additionally obtained where dense non-Lambertian reflectance appears, making more potential applications feasible, such as digital relighting and material classification.

The rest of the paper is organized as follows. The next section presents the problem formulation of colour PS incorporating DRM and the relevant PS methods. Section 3 first overviews the flow and original contribution of the proposed colour DRM-based PS method and then, elucidates the specific two steps in the subsequent subsections. It follows the comprehensive evaluations on surface orientation estimation using both synthetic and real images in Section 4, while the last section summarizes the conclusions and proposes the future works.

## 2. Colour PS Incorporating Dichromatic Reflectance Model

### 2.1. Generic Colour PS Problem Formulation

Figure 1 shows the schematic diagram of the hardware components of the sensor and the necessary data for the generic colour PS. The hardware components are an RGB camera and a set of directional point sources whose positions are priorly known. Assume that ambient light is completely blocked, the colour PS requires that the RGB camera captures a colour image for the target surface with one source lit at a time. Given the *N* colour images illuminated under *N* different sources, the objective of the colour PS is to identify surface orientations pixel-wise. More specifically, given the colour image irradiance measurements at pixel $\left(i,j\right)$, ${\left\{e_{k}^{i,j*}\right\}}_{k\in \left[1,N\right]}$, under the unit illumination directions, ${\left\{{\bar{l}}_{k}^{i,j}\right\}}_{k\in \left[1,N\right]}$, estimated from the light positions, the colour PS aims to derive the unit surface normal, ${\bar{n}}^{i,j}$, by minimizing the sum of *N* corresponding square residuals:(1)$$\epsilon^{i,j}=\sum_{k=1}^{N}{\left[{\left(e_{k}^{i,j*}\right)}^{T}w-f\left({\bar{n}}^{i,j},{\bar{l}}_{k}^{i,j}\right)\right]}^{2}\to \min_{{\bar{n}}^{i,j}}.$$
where w is a vector of weights for colour channels whereas *f* is a forward function of ${\bar{n}}^{i,j}$ having ${\bar{l}}_{k}^{i,j}$ as parameters, which corresponds to the weighted measurements. Note that $\bar{\left(\right)}$ and ${\left(\right)}^{*}$ mean the unit version and the measurement of () respectively throughout the paper.

### 2.2. Colour Image Formation Model

Figure 2 shows the coordinate setups that are necessary for the colour PS to model the imaging geometry and the light configuration. The coordinate frames are the 3D camera frame, {C}, the 2D pixel frame, {P}, and the world frame, {W}, where its XY plane is known and referred to as the reference plane. These coordinate frames follow the convention given in [24]. A 3D surface point, ${}^{\left\{C\right\}}\tilde{X}$, is projected to be a 2D image point, ${}^{\left\{P\right\}}x$, in the RGB camera using the perspective projection as:(2)$${}^{\left\{P\right\}}x=\frac{1}{{}^{\left\{C\right\}}Z}M_{p}\hat{K}{}^{\left\{C\right\}}\tilde{X}=\frac{1}{{}^{\left\{C\right\}}Z}\left[\begin{array}{ccc} m_{x} & 0 & o_{x}\\ 0 & m_{y} & o_{y}\\ 0 & 0 & 1 \end{array}\right]\left[\begin{array}{ccc} f & 0 & 0\\ 0 & f & 0\\ 0 & 0 & 1 \end{array}\right]{}^{\left\{C\right\}}\tilde{X},$$
where $m_{x}$ and $m_{y}$ are each the number of pixels per unit distance along *x* and *y* direction respectively. $o_{x}$ and $o_{y}$ are the principal point location in {P}, and *f* is the effective focal length. Note that ${}^{\left\{P\right\}}x={\left[{}^{\left\{P\right\}}x,{}^{\left\{P\right\}}y,1\right]}^{⊤}$ and ${}^{\left\{C\right\}}\tilde{X}={\left[{}^{\left\{C\right\}}X,{}^{\left\{C\right\}}Y,{}^{\left\{C\right\}}Z\right]}^{⊤}$. ${}^{\left\{P\right\}}x$ is rasterized into a regular grid in the camera as:(3)$$\left[\begin{array}{c} j\\ i \end{array}\right]=\mathrm{ceil}\left(\left[\begin{array}{c} {}^{\left\{P\right\}}x+0.5\\ {}^{\left\{P\right\}}y+0.5 \end{array}\right]\right),$$
where ceil(·) denotes the ceiling operator. {W} and {C} are related with a 3D rigid transformation: ${}^{\left\{C\right\}}\tilde{X}=R_{W}{}^{\left\{W\right\}}\tilde{X}+t_{W}$.

Let the *k*th light position be ${}^{\left\{C\right\}}{\tilde{S}}_{k}={\left[{}^{\left\{C\right\}}X_{k},{}^{\left\{C\right\}}Y_{k},{}^{\left\{C\right\}}Z_{k}\right]}^{⊤}$. ${}^{\left\{C\right\}}{\tilde{S}}_{k}$ is associated in the spherical space as:(4)$$\left\{\begin{array}{c} {}^{\left\{C\right\}}X_{k}={}^{\left\{C\right\}}r_{k}\sin{}^{\left\{C\right\}}\theta_{k}\cos{}^{\left\{C\right\}}\phi_{k}\\ {}^{\left\{C\right\}}Y_{k}={}^{\left\{C\right\}}r_{k}\sin{}^{\left\{C\right\}}\theta_{k}\sin{}^{\left\{C\right\}}\phi_{k}\\ {}^{\left\{C\right\}}Z_{k}={}^{\left\{C\right\}}r_{k}\cos{}^{\left\{C\right\}}\theta_{k}, \end{array}\right.$$
where ${}^{\left\{C\right\}}r_{k}$, ${}^{\left\{C\right\}}\theta_{k}$ and ${}^{\left\{C\right\}}\phi_{k}$ are the radius, zenith and azimuth angle, respectively. The *N* lights are configured such that they have the same zenith angle and radius, and uniformly distributed azimuth angles around the optical axis, i.e., ${}^{\left\{C\right\}}\theta_{k}=\theta$, ${}^{\left\{C\right\}}r_{k}=r$ and ${}^{\left\{C\right\}}\phi_{k}=\frac{360^{\circ }\left(k-1\right)}{N}+\phi_{0}$.

Having established the imaging geometry and light configuration, the imaging photometry is then modeled using the physically-based DRM [22]. The DRM properly characterizes reflectance for dielectric materials [25] and is represented by a linear combination of the diffuse and specular components. Assume that pixel $\left(i,j\right)$ is shadow-free, the image irradiance for colour channel *c* after light strength normalisation and vignetting correction is written as:(5)$$e_{c}^{i,j}=e_{d,c}^{i,j}+e_{s,c}^{i,j}=d_{c}^{i,j}f_{d}^{i,j}+s_{c}f_{s}^{i,j},$$
where $e_{d,c}^{i,j}$ and $e_{s,c}^{i,j}$ are the diffuse and specular components, and $d_{c}^{i,j}$ and $s_{c}$ are diffuse and specular colours which are wavelength-dependent and represented by:
(6)$$\left\{\begin{array}{c} d_{c}^{i,j}=\int_{0}^{\infty }\Phi \left(\lambda \right)R^{i,j}\left(\lambda \right)C_{c}\left(\lambda \right)d\lambda ,\\ S_{c}=\int_{0}^{\infty }\Phi \left(\lambda \right)C_{c}\left(\lambda \right)d\lambda . \end{array}\right.$$

In the equation, λ is the wavelength, $\Phi \left(\lambda \right)$ is the source spectral power density (SPD), $R^{i,j}\left(\lambda \right)$ is the spectral body reflectance, and $C_{c}\left(\lambda \right)$ is the camera spectral sensitivity for the colour channel. $f_{d}^{i,j}$ and $f_{s}^{i,j}$ represent the diffuse and specular geometrical scaling factors that are characterized by BRDF.

Since the scaling factors are illumination-dependent, let the image irradiance Equation (Equation 5) generalized in the RGB colour space be with the *k*th illumination as:(7)$$e_{k}^{i,j}=e_{d,k}^{i,j}+e_{s,k}^{i,j}={\bar{d}}_{RGB}^{i,j}f_{d,k}^{i,j}+{\bar{s}}_{RGB}f_{s,k}^{i,j}=\left[\begin{array}{c} {\bar{d}}_{R}^{i,j}\\ {\bar{d}}_{G}^{i,j}\\ {\bar{d}}_{B}^{i,j} \end{array}\right]f_{d,k}^{i,j}+\left[\begin{array}{c} {\bar{s}}_{R}\\ {\bar{s}}_{G}\\ {\bar{s}}_{B} \end{array}\right]f_{s,k}^{i,j},$$
where ${\bar{d}}_{RGB}^{i,j}={\left\{d_{c}^{i,j}\right\}}_{c=R,G,B}$ and ${\bar{s}}_{RGB}={\left\{s_{c}\right\}}_{c=R,G,B}$ are the unit diffuse and specular colour, respectively. The diffuse geometrical scaling factor is characterized with the *k*th unit illumination direction ${\bar{l}}_{k}^{i,j}$ by Lambertian model as:(8)$$f_{d,k}^{i,j}=k_{d}^{i,j}{\left({\bar{n}}^{i,j}\right)}^{⊤}{\bar{l}}_{k}^{i,j},$$
where $k_{d}^{i,j}$ is the diffuse reflectance factor. Since the surface deviation is small compared with the light-surface distance, ${\bar{l}}_{k}^{i,j}$ can be approximately derived using a vector from the point intersected by ray $\left(i,j\right)$ and the reference plane to ${}^{\left\{C\right\}}{\tilde{S}}_{k}$. One of the common options of the specular geometrical scaling factor, on the other hand, is a model using the Blinn-Phong BRDF [26] because of its computational efficiency and high performance in characterizing a wide range of isotropic reflectance [27]:(9)$$f_{s,k}^{i,j}=k_{s}^{i,j}{\left({\left({\bar{n}}^{i,j}\right)}^{T}{\bar{h}}_{k}^{i,j}\right)}^{\beta^{i,j}},$$
where $k_{s}^{i,j}$ is the specular reflectance factor, ${\bar{h}}_{k}^{i,j}$ is the unit half vector, and $\beta^{i,j}$ is the shininess coefficient. The unit half vector is written as:(10)$${\bar{h}}_{k}^{i,j}=\frac{{\bar{l}}_{k}^{i,j}+{\bar{v}}^{i,j}}{∥{\bar{l}}_{k}^{i,j}+{\bar{v}}^{i,j}∥},$$
where ${\bar{v}}^{i,j}$ is the unit viewer direction and determined by:
(11)$$\left\{\begin{array}{c} v^{i,j}=-{\left[\frac{j-o_{x}}{m_{x}},\frac{i-o_{y}}{m_{y}},f\right]}^{𐊗},\\ {\bar{v}}^{i,j}=v^{i,j}/\|v^{i,j}\|. \end{array}\right.$$


### 2.3. Colour PS Methods Using DRM

Existing colour PS methods using DRM contain two major variants, representative works of which were developed by Barsky et al. [21] and by Zickler et al. [23]. The PS method of Barsky finds the unit surface normal ${\bar{n}}^{i,j}$ and the diffuse reflectance factor $k_{d}^{i,j}$ such that the residual between the colour image irradiances weighted by ${\bar{d}}_{RGB}^{i,j}$ and the diffuse geometrical scaling factor is minimised:(12)$$\epsilon^{i,j}=\sum_{k=1}^{N_{b}}{\left[{\left(e_{k}^{i,j*}\right)}^{⊤}{\bar{d}}_{RGB}^{i,j}-f_{d,k}^{i,j}\right]}^{2}=\sum_{k=1}^{N_{b}}{\left[{\left(e_{k}^{i,j*}\right)}^{⊤}{\bar{d}}_{RGB}^{i,j}-k_{d}^{i,j}{\left({\bar{n}}^{i,j}\right)}^{⊤}{\bar{l}}_{k}^{i,j}\right]}^{2}\to \min_{{\bar{n}}^{i,j},k_{d}^{i,j}},$$
where $N_{b}\left(\leq N\right)$ is the number of specular-free and shadow-free images. The reason for the use of $N_{b}$ instead of *N* is that the Barsky’s PS method assumes that non-Lambertian reflectance appears sparsely among observations, which should be rejected as outliers. As the first process for outlier rejection, the direct principal component analysis (DPCA) [28] on ${\left[E_{RGB}^{i,j*}\right]}^{⊤}\left[E_{RGB}^{i,j*}\right]$ is performed to estimate ${\bar{d}}_{RGB}^{i,j}$, where $\left[E_{RGB}^{i,j*}\right]={\left[e_{1}^{i,j*},e_{2}^{i,j*},\ldots ,e_{N}^{i,j*}\right]}^{⊤}$. Given ${\bar{s}}_{RGB}$ and ${\bar{d}}_{RGB}^{i,j}$, the method then determines $f_{s,k}^{i,j}$ as:(13)$$f_{s,k}^{i,j}=\frac{{\left(e_{k}^{i,j*}\right)}^{T}{\bar{s}}_{RGB}-\left({\left(e_{k}^{i,j*}\right)}^{⊤}{\bar{d}}_{RGB}^{i,j}\right)\left({\left({\bar{d}}_{RGB}^{i,j}\right)}^{⊤}{\bar{s}}_{RGB}\right)}{1-{\left({\left({\bar{d}}_{RGB}^{i,j}\right)}^{T}{\bar{s}}_{RGB}\right)}^{2}}.$$

A threshold on $f_{s,k}^{i,j}$ is finally used to determine pixels to be rejected as outliers, and this subsequently determines $N_{b}$.

The objective function (Equation 12) of this PS method is not different from that of the conventional Lambertian-based PS method [3]. Given ${\left\{{\bar{l}}_{k}^{i,j}\right\}}_{k\in \left[1,M\right]}$*a priori*, ${\bar{n}}^{i,j}$ together with $k_{d}^{i,j}$, which minimizes the objective function (Equation 12), is analytically determined as:$$\left\{\begin{array}{c} k_{d}^{i,j}=∥{\left({\left[L^{i,j}\right]}^{⊤}\left[L^{i,j}\right]\right)}^{-1}{\left[L^{i,j}\right]}^{⊤}\left[E_{RGB}^{i,j*}\right]{\bar{d}}_{RGB}^{i,j}∥,\\ {\bar{n}}^{i,j}=\frac{{\left({\left[L^{i,j}\right]}^{⊤}\left[L^{i,j}\right]\right)}^{-1}{\left[L^{i,j}\right]}^{⊤}\left[E_{RGB}^{i,j*}\right]{\bar{d}}_{RGB}^{i,j}}{k_{d}^{i,j}}, \end{array}\right.$$
where $\left[L^{i,j}\right]={\left[{\bar{l}}_{1}^{i,j},{\bar{l}}_{2}^{i,j},\ldots ,{\bar{l}}_{N_{b}}^{i,j}\right]}^{⊤}$ and $\mathrm{rank}(\left[L^{i,j}\right])=3$. While the effect of specularity rejection through the use of ${\bar{s}}_{RGB}$ is exhibited, the estimation of ${\bar{d}}_{RGB}^{i,j}$ using DPCA is erroneous due to the existence of specularities. Furthermore, all the specularities are rejected as outliers though useful information may be contained.

Unlike Barsky’s PS method, which performs PS in the RGB colour space, the PS method of Zickler estimates surface orientations in the SUV colour space, or the UV colour space as S is not considered:(14)$$\begin{array}{c} \epsilon^{i,j}=\sum_{k=1}^{N}{\left[{\left(e_{UV,k}^{i,j*}\right)}^{T}{\bar{d}}_{UV}^{i,j}-\alpha^{i,j}{\left({\bar{n}}^{i,j}\right)}^{T}{\bar{l}}_{k}^{i,j}\right]}^{2}\to \min_{{\bar{n}}^{i,j}}, \end{array}$$
where $\alpha^{i,j}$ is a scaling factor. The UV colour subspace includes more observations than the RGB colour space in surface orientation estimation. To operate in the SUV space, the colour image irradiance is transformed from RGB to SUV as:(15)$$e_{SUV,k}^{i,j}≜e_{k}^{i,j}=R_{s}e_{k}^{i,j}={\bar{d}}_{SUV}^{i,j}f_{d,k}^{i,j}+{\bar{s}}_{SUV}f_{s,k}^{i,j},$$
where ${\bar{d}}_{SUV}^{i,j}={\left[{\bar{d}}_{U}^{i,j},{\bar{d}}_{V}^{i,j},{\bar{d}}_{S}^{i,j}\right]}^{T}$, ${\bar{s}}_{SUV}={\left[0,0,1\right]}^{T}$, and $R_{s}\in SO\left(3\right)$ is any transformation that yields ${\bar{s}}_{SUV}=R_{s}{\bar{s}}_{RGB}={\left[0,0,1\right]}^{T}$. UV forms a specular-free colour subspace that is invariant to $f_{s,k}^{i,j}$. The Zickler’s PS method first estimates ${\bar{d}}_{UV}^{i,j}$ of unit length using DPCA on ${\left[E_{UV}^{i,j*}\right]}^{⊤}\left[E_{UV}^{i,j*}\right]$. It then becomes the conventional Lambertian-based PS problem and the solution is obtained in the similar form to Equation (Equation 14). Compared with the Barsky’s PS, the Zickler’s PS utilizes more observations and advantageously performs PS in the specular-free UV colour subspace. Shape information from specularities is however neglected again, and the method can only provide diffuse component up to the normalised RGB space. The diffuse-specular separation in the RGB space remains open.

## 3. DRM-Based Colour PS Method Using Diffuse-Specular Separation

### 3.1. Overview of the Proposed Color PS Method

Figure 3 shows the flow and original contributions of the proposed DRM-based colour PS method using the diffuse-specular separation. The proposed method consists of two processes: Step 1, which is diffuse-specular separation using a PS, and Step 2, which is the surface normal refinement. Step 1 identifies good approximate surface normals, whereas Step 2 further refines and finalizes them with maximum accuracy. The steps are marked by red boxes.

Step 1 is composed of four sub-processes: diffuse colour estimation, diffuse-specular separation in RGB space, PS in UV space and the outlier estimation. After completing shadow rejection as a preprocess, and generating the shadow-free image irradiance matrix $\left[{\underline{E}}_{RGB}^{i,j*}\right]$, Step 1 begins with the diffuse colour estimation using the robust principal component analysis (RPCA) proposed in this paper and derives ${\bar{d}}_{RGB}^{i,j}$. Note that $\underline{\left(\right)}$ is a quantity of () after shadow rejection. The diffuse color estimation also outputs a specularity map, which describes the distribution of specular reflection. The diffuse-specular separability is then checked for each pixel by deriving the diffuse-specular chromatic angle $\psi^{i,j}$ in RGB space. For separable pixels, the proposed method performs PS in the specular-free UV colour subspace similarly to Zickler’s PS method and derives the initial guess of the unit surface normal ${\left({\bar{n}}^{i,j}\right)}_{1}$ and the diffuse reflectance factor ${\left(k_{d}^{i,j}\right)}_{1}$ similarly to Barsky’s PS. ${\left(\right)}_{1}$ means the initial guess of (). In parallel, the specular geometrical scaling factors, ${\left({\underline{f}}_{s}^{i,j}\right)}_{1}$, are derived through the diffuse-specular separation process using Equation (Equation 13) for the subsequent surface normal refinement. Observations not in the specularity map are clamped to zero in ${\left({\underline{f}}_{s}^{i,j}\right)}_{1}$. The surface normals and the other parameters are then refined by fitting the nonlinear DRM using LMA with regularisation.

In Step 2, the initial guess of the specular parameters, ${\left(k_{s}^{i,j}\right)}_{1}$ and ${\left(\beta^{i,j}\right)}_{1}$, are drived from ${\left({\bar{n}}^{i,j}\right)}_{1}$ and ${\left({\underline{f}}_{s}^{i,j}\right)}_{1}$. Finally, PS for the surface normal refinement derives ${\bar{n}}^{i,j}$ and $k_{d}^{i,j}$ with enhanced accuracy while additionally updating $k_{s}^{i,j}$ and $\beta^{i,j}$. The strength of the proposed method lies in this PS with the diffuse-specular separation. For pixels where sparse non-Lambertian reflectances appear, the first step is sufficient to reliably estimate the surface orientations. To tackle with more general problems with dense non-Lambertian reflectances, the second step is additionally required to exploit surface normal information from specularities in the DRM with the designed parameter initialisation strategy. The rest of this section elucidates the proposed method in detail.

### 3.2. Diffuse-Specular Separation

#### 3.2.1. Diffuse Color Estimation

The proposed RPCA derives ${\bar{d}}_{RGB}^{i,j}$ and the specularity map as follows:1.Perform principle component analysis (PCA) for ${\left[{\underline{E}}_{RGB}^{i,j*}\right]}^{⊤}\left[{\underline{E}}_{RGB}^{i,j*}\right]$ to estimate ${\bar{d}}_{RGB}^{i,j}$;2.Compute residual matrix: $\left[R_{d}^{i,j}\right]=\left[{\underline{E}}_{RGB}^{i,j*}\right]{\bar{d}}_{RGB}^{i,j}{\left({\bar{d}}_{RGB}^{i,j}\right)}^{⊤}-\left[{\underline{E}}_{RGB}^{i,j*}\right]$.3.Compute residual vector: $r_{d}^{i,j}=\sqrt{{\left(r_{d,1}^{i,j}\right)}^{\circ 2}+{\left(r_{d,2}^{i,j}\right)}^{\circ 2}+{\left(r_{d,3}^{i,j}\right)}^{\circ 2}}$, where $\left[R_{d}^{i,j}\right]=\left[r_{d,1}^{i,j},r_{d,2}^{i,j},r_{d,3}^{i,j}\right]$ and ∘2 denotes the hadamard square.4.If the mean of $r_{d}^{i,j}$, $\mathrm{mean}(r_{d}^{i,j})$, is smaller than the threshold $T_{d}$, terminate RPCA and output the current estimate of ${\bar{d}}_{RGB}^{i,j}$ and the specularity map. Otherwise, find the element that provides the maximal value of $\left(\frac{r_{d}^{i,j}-\mathrm{mean}\left(r_{d}^{i,j}\right)}{\mathrm{std}(r_{d}^{i,j})}\right)$ and register (i,j) in the specularity map. Remove the corresponding row vector in $\left[{\underline{E}}_{RGB}^{i,j*}\right]$ and repeat from step 1.

The RPCA eliminates image irradiances that are inappropriate for diffuse colour estimation.

The validity of the RPCA can be explained as follows. The shadow-free image irradiance matrix $\left[{\underline{E}}_{RGB}^{i,j*}\right]$ is expanded from Equation (Equation 7) and decomposed as
(16)$$\begin{array}{ccc} \left[{\underline{E}}_{RGB}^{i,j*}\right] & = & {\underline{f}}_{d}^{i,j}{\left({\bar{d}}_{RGB}^{i,j}\right)}^{⊤}+{\underline{f}}_{s}^{i,j}{\left({\bar{s}}_{RGB}\right)}^{⊤}\\ & = & \left[\begin{array}{c} f_{d,1}^{i,j}\\ f_{d,2}^{i,j}\\ \ldots \\ f_{d,N_{p}}^{i,j} \end{array}\right]\left[\begin{array}{ccc} {\bar{d}}_{R}^{i,j} & {\bar{d}}_{G}^{i,j} & {\bar{d}}_{B}^{i,j} \end{array}\right]+\left[\begin{array}{c} f_{s,1}^{i,j}\\ f_{s,2}^{i,j}\\ \ldots \\ f_{s,N_{p}}^{i,j} \end{array}\right]\left[\begin{array}{ccc} {\bar{s}}_{R} & {\bar{s}}_{G} & {\bar{s}}_{B} \end{array}\right] \end{array}$$
where most entries in ${\underline{f}}_{s}^{i,j}$ are near-zero since images are often near specular-free. In such a case, a non-zero residual occurs if ${\underline{f}}_{s}^{i,j}$ contains non-negligible entries. The RPCA algorithms robustly and accurately estimate ${\bar{d}}_{RGB}^{i,j}$ by keeping rejecting specularities until the residual becomes smaller than $T_{d}$. Unlike Barsky’s PS method, which identifies specularities after ${\bar{d}}_{RGB}^{i,j}$ estimation, the proposed algorithms derive ${\bar{d}}_{RGB}^{i,j}$ and the specularity map simultaneously.

While the originality and superiority of the proposed RPCA algorithms have been identified, the performance of the RPCA algorithms depends on the value of $T_{d}$. If $T_{d}$ is too large, the RPCA algorithms are tolerant to specularities and perform similarly to the DPCA algorithms. If $T_{d}$ is too small, the RPCA algorithms reject innocent pixels as specularities and make images sensitive to noise.

#### 3.2.2. Diffuse-Specular Separability Check

Having ${\bar{d}}_{RGB}^{i,j}$ estimated, the diffuse-specular chromatic angle, used to perform the separability check, is derived as
(17)$$\cos\psi^{i,j}={\left({\bar{d}}_{RGB}^{i,j}\right)}^{T}{\bar{s}}_{RGB},$$

Asymptotically, the chromatic angle gives
(18)$$\lim_{\psi^{i,j}\to 0}{\bar{d}}_{RGB}^{i,j}={\bar{s}}_{RGB}$$

This means that $\psi^{i,j}$ should be sufficiently larger than 0 in order for ${\bar{d}}_{RGB}^{i,j}$ and ${\bar{s}}_{RGB}$ to be distinct and separable. The threshold of the chromatic angle to determine the separability is $T_{c}$. If $\psi^{i,j}\geq T_{c}$, ${\bar{d}}_{RGB}^{i,j}$ and ${\bar{s}}_{RGB}$ are sufficiently distinct and pixel $\left(i,j\right)$ is considered as separable. Similarly to $T_{d}$, $T_{c}$ should be chosen carefully. A small $T_{c}$ results in yielding false separation. Once ${\bar{d}}_{RGB}^{i,j}$ and ${\bar{s}}_{RGB}$ have been identified, ${\left({\underline{f}}_{s}^{i,j}\right)}_{1}$ is then initialised using Equation (Equation 13).

#### 3.2.3. PS in UV Space

Since the PS is performed in the UV colour space, the unit diffuse colour ${\bar{d}}_{RGB}^{i,j}$ should be converted into that in UV colour space ${\bar{d}}_{UV}^{i,j}$. The PS in UV color space allows more observations as described in the Zickler’s PS method. With $R_{s}$, the estimated ${\bar{d}}_{RGB}^{i,j}$ is transformed to the SUV space as ${\hat{d}}^{i,j}={\left[{\hat{d}}_{U}^{i,j},{\hat{d}}_{V}^{i,j},{\hat{d}}_{S}^{i,j}\right]}^{⊤}$. The diffuse color in UV space ${\hat{d}}_{UV}^{i,j}$ is then given by
(19)$${\hat{d}}_{UV}^{i,j}={\left[{\hat{d}}_{U}^{i,j},{\hat{d}}_{V}^{i,j}\right]}^{⊤}=\kappa_{z}^{i,j}{\bar{d}}_{UV}^{i,j}$$
where ${\bar{d}}_{UV}^{i,j}={\left[{\bar{d}}_{U}^{i,j},{\bar{d}}_{V}^{i,j}\right]}^{⊤}$ is a unit vector, and
(20)$$\kappa_{z}^{i,j}=\sqrt{{\left({\hat{d}}_{U}^{i,j}\right)}^{2}+{\left({\hat{d}}_{V}^{i,j}\right)}^{2}}>0.$$

Image irradiances of separable pixels are also transformed to the SUV colour space as $\left[{\underline{E}}_{SUV}^{i,j}\right]$ using the same $R_{s}$. $\left[{\underline{E}}_{UV}^{i,j}\right]$ is then formed by picking the first two columns of $\left[{\underline{E}}_{SUV}^{i,j}\right]$. Due to the presence of image noise, the PS in UV space is modified from Equation (Equation 15) to reject noise-corrupted image irradiances as outliers using studentised residuals [29]:(21)$$\begin{array}{ccc} \epsilon^{i,j} & = & \sum_{k=1}^{N_{p}}{\left[{\left(e_{UV,k}^{i,j*}\right)}^{⊤}{\bar{d}}_{UV}^{i,j}-{\left(\kappa_{z}^{i,j}\right)}^{2}f_{d,k}^{i,j}\right]}^{2}\\ & = & \sum_{k=1}^{N_{p}}{\left[{\left(e_{UV,k}^{i,j*}\right)}^{⊤}{\bar{d}}_{UV}^{i,j}-{\left(\kappa_{z}^{i,j}\right)}^{2}{\left(k_{d}^{i,j}\right)}_{1}{\left({\bar{n}}^{i,j}\right)}_{1}^{⊤}{\bar{l}}_{k}^{i,j}\right]}^{2}\\ & \to & \min_{{\left({\bar{n}}^{i,j}\right)}_{1},{\left(k_{d}^{i,j}\right)}_{1}}, \end{array}$$
where $N_{p}$ is the number of shadow-free images, so $N_{b}\leq N_{p}\leq N$. The derivation of ${\left(k_{d}^{i,j}\right)}_{1}$ in addition to ${\left({\bar{n}}^{i,j}\right)}_{1}$ similarly to the Barsky’s PS method enables full recovery of the colour diffuse component. $N_{p}$ is used instead of *N* and $N_{b}$ because shadow-free images are effective for reliability.

After the PS, ${\bar{n}}^{i,j}$ and $k_{d}^{i,j}$ are initialised as ${\left({\bar{n}}^{i,j}\right)}_{1}$ and ${\left(k_{d}^{i,j}\right)}_{1}$ respectively. The diffuse component matrix, $\left[E_{d,RGB}^{i,j}\right]$, is then re-rendered as:(22)$$\left[E_{d,RGB}^{i,j}\right]=\max\left({\left(k_{d}^{i,j}\right)}_{1}\left[L^{i,j}\right]{\left({\bar{n}}^{i,j}\right)}_{1},0_{N}\right){\left({\bar{d}}_{RGB}^{i,j}\right)}^{T},$$
where $0_{N}$ is an N×1 vector with all zero entries. The specular component matrix, $\left[E_{s,RGB}^{i,j}\right]$, is obtained by $\max\left({\left({\underline{f}}_{s}^{i,j}\right)}_{1}{\bar{s}}_{RGB}^{T},0_{N}\right)$. With the separated colour diffuse and specular components, the proposed method allows the functionality of specular removal and intrinsic image decomposition.

#### 3.2.4. Outlier Estimation

Let the residual vector ${\underline{r}}^{i,j}$ be written as:(23)$${\underline{r}}^{i,j}=\left[{\underline{E}}_{UV}^{i,j}\right]{\bar{d}}_{UV}^{i,j}-\left[{\underline{L}}^{i,j}\right]n^{i,j}=\left[{\underline{E}}_{UV}^{i,j}\right]{\bar{d}}_{UV}^{i,j}-\left[{\underline{L}}^{i,j}\right]\rho^{i,j}{\bar{n}}^{i,j},$$
where $n^{i,j}=\rho^{i,j}{\bar{n}}^{i,j}$ is the scaled normal, $\left[{\underline{L}}^{i,j}\right]={\left[{\bar{l}}_{1}^{i,j},{\bar{l}}_{2}^{i,j},\ldots ,{\bar{l}}_{M}^{i,j}\right]}^{T}$, and $\rho^{i,j}=k_{d}^{i,j}\kappa_{z}^{i,j}$. The hat matrix $\left[H_{a}^{i,j}\right]$ [29] is represented by:(24)$$\left[{\underline{H}}_{a}^{i,j}\right]=\left[{\underline{L}}^{i,j}\right]{\left({\left[{\underline{L}}^{i,j}\right]}^{T}\left[{\underline{L}}^{i,j}\right]\right)}^{-1}{\left[{\underline{L}}^{i,j}\right]}^{T}.$$

Diagonal entries in $\left[{\underline{H}}_{a}^{i,j}\right]$ are leverages and the *k*th leverage is denoted as $h_{a,k}^{i,j}$. These leverages quantify the influence that the observed $\left[{\underline{E}}_{UV}^{i,j}\right]{\bar{d}}_{UV}^{i,j}$ on their predicted values $\left[{\underline{L}}^{i,j}\right]n^{i,j}$. Each entry in the studentized residual vector ${\tilde{\underline{r}}}^{i,j}$ is then approximated using:(25)$${\tilde{r}}_{k}^{i,j}=\frac{r_{k}^{i,j}}{\sqrt{{\left(MSE\right)}^{i,j}\left(1-h_{a,k}^{i,j}\right)}},$$
where ${\left(\mathrm{MSE}\right)}^{i,j}$ represents the mean squared error of ${\underline{r}}^{i,j}$. If any entry in $|{\tilde{\underline{r}}}^{i,j}|$ exceeds $T_{o}$, this entry is estimated as outliers. The proposed PS rejects outliers until no more can be detected or ${\left(\mathrm{MSE}\right)}^{i,j}$ is smaller than a tolerance, $T_{m}$. With the general rule of thumb in detecting outliers using studentised residuals, $T_{o}$ and $T_{m}$ can be chosen as 2.5 and $9\sigma_{n}^{2}$, where $\sigma_{n}$ is the standard deviation of additive white Gaussian noise that can be estimated using the method given by [30].

### 3.3. Surface Normal Refinement

#### 3.3.1. Specular Parameter Initialisation

If dense non-Lambertian reflectance appears at pixel $\left(i,j\right)$, which implies that more than one entry in ${\left({\underline{f}}_{s}^{i,j}\right)}_{1}$ are in the specularity map, surface normal refinement is necessary since the sparse non-Lambertian reflectance assumption in the diffuse-specular separation is violated. With the estimated ${\left({\underline{f}}_{s}^{i,j}\right)}_{1}$, the next objective is to initialise the specular parameters, ${\left(k_{s}^{i,j}\right)}_{1}$ and ${\left(\beta^{i,j}\right)}_{1}$. The specific cost functional is given by:(26)$$\epsilon_{s}^{i,j}=\sum_{k=1}^{N_{q}}{\left({\left(f_{s,k}^{i,j}\right)}_{1}-{\left(k_{s}^{i,j}\right)}_{1}{\left({\left({\bar{h}}_{k}^{i,j}\right)}^{⊤}{\left({\bar{n}}^{i,j}\right)}_{1}\right)}^{{\left(\beta^{i,j}\right)}_{1}}\right)}^{2}\to \min_{{\left(k_{s}^{i,j}\right)}_{1},{\left(\beta^{i,j}\right)}_{1}},$$
where $N_{q}$ is the number of specularity maps showing pixel $\left(i,j\right)$ and $N_{q}\geq 2$. Equation (Equation 26) suggests a nonlinear least-squares problem, while it can be manipulated to the natural logarithm domain as a linear problem given by:(27)$$\epsilon_{ls}^{i,j}=\sum_{k=1}^{N_{q}}{\left(\ln{\left(f_{s,k}^{i,j}\right)}_{1}-\ln{\left(k_{s}^{i,j}\right)}_{1}-{\left(\beta^{i,j}\right)}_{1}\ln\left({\left({\bar{h}}_{k}^{i,j}\right)}^{⊤}{\left({\bar{n}}^{i,j}\right)}_{1}\right)\right)}^{2}\to \min_{{\left(k_{s}^{i,j}\right)}_{1},{\left(\beta^{i,j}\right)}_{1}}.$$

Let ${\left({\underline{\underline{f}}}_{s}^{i,j}\right)}_{1}$ with $N_{q}$ rows consist of entries in ${\left({\underline{f}}_{s}^{i,j}\right)}_{1}$ in the specularity map and ${\left[\underline{\underline{H}}\right]}^{i,j}={\left[{\bar{h}}_{1}^{i,j},{\bar{h}}_{2}^{i,j},...,{\bar{h}}_{N_{q}}^{i,j}\right]}^{T}$ be the matrix comprised of the corresponding unit half vectors. Then, the solution of $\ln{\left(k_{s}^{i,j}\right)}_{1}$ and ${\left(\beta^{i,j}\right)}_{1}$ to Equation (Equation 27) is given analytically by:(28)$$\left[\begin{array}{c} {\left(\beta^{i,j}\right)}_{1}\\ \ln{\left(k_{s}^{i,j}\right)}_{1} \end{array}\right]={\left({\left[{\underline{\underline{S}}}^{i,j}\right]}^{⊤}\left[{\underline{\underline{S}}}^{i,j}\right]\right)}^{-1}{\left[{\underline{\underline{S}}}^{i,j}\right]}^{⊤}\ln{\left({\underline{\underline{f}}}_{s}^{i,j}\right)}_{1},$$
where
(29)$$\left[{\underline{\underline{S}}}^{i,j}\right]=\left[\ln\left({\left[\underline{\underline{H}}\right]}^{i,j}{\left({\bar{n}}^{i,j}\right)}_{1}\right),1_{N_{q}}\right]$$
and $1_{N_{q}}$ is a $N_{q}\times 1$ vector with all entries equal to 1.

#### 3.3.2. Surface Normal Refinement in DRM

All parameters in the DRM have been initialised up to this point. The LMA is then employed to iteratively refine these parameters by solving the optimization problem:(30)$$\begin{array}{c} \epsilon^{i,j}=\sum_{k=1}^{N}\left({\left(e_{k}^{i,j}\right)}^{⊤}{\bar{s}}_{RGB}-k_{d}^{i,j}{\left({\bar{n}}^{i,j}\right)}^{⊤}{\bar{l}}_{k}^{i,j}{\left({\bar{d}}_{RGB}^{i,j}\right)}^{⊤}{\bar{s}}_{RGB}-k_{s}^{i,j}{\left({\left({\bar{n}}^{i,j}\right)}^{⊤}{\bar{h}}_{k}^{i,j}\right)}^{\beta^{i,j}}\right.\\ {\left.+T_{\alpha }\left(1-{\left({\bar{n}}^{i,j}\right)}^{⊤}{\bar{n}}^{i,j}\right)\right)}^{2}\to \min_{{\bar{n}}^{i,j},k_{d}^{i,j},k_{s}^{i,j},\beta^{i,j}}, \end{array}$$
where the first three terms are from the generalised formulation, and, the fourth regularization term with constant $T_{\alpha }$ is added. The regularisation term reduces the change of surface normal refinement from the initial guess of ${\left({\bar{n}}^{i,j}\right)}_{1}$ and prevents overfitting. Larger value of $T_{\alpha }$ strengthens the robustness but limits the capability of surface normal refinement.

The strength of the proposed PS method lies in the determination of the surface normals while separating the diffuse and specular components in RGB space, eliminating outliers and then tuning the diffuse and specular reflectance factors simultaneously. The determination considering diffuse and specular reflections simultaneously makes the surface normal estimation more accurately. In addition, the proposed method can be used for specular removal [31] and intrinsic image decomposition [32]. Parameters characterizing $f_{s,k}$ are determined at pixels, so the proposed method could also be used for digital relighting and material classification where dense non-Lambertian reflectance appears.

## 4. Performance Evaluation on Surface Orientation Estimation

### 4.1. Evaluations Using Synthetic Images

The first experiment aims to evaluate the effectiveness of surface normal refinement for dense non-Lambertian reflectance using synthetic input images. A scene with six different-coloured spheres were rendered under 32 illuminants using the Blinn-Phong model. The sphere was adopted as the scene geometry since it samples all the possibilities of surface orientations for the visible surface. Different colours of the spheres were introduced for more comprehensive evaluations of the proposed PS method on surfaces with various spectral reflectances. The 32 illuminants were chosen to provide a sufficient number of pixels with dense non-Lambertian reflectance for evaluation. ${\bar{s}}_{RGB}$ was set at ${\left[0.5774,0.5774,0.5774\right]}^{T}$. Three colours of the spheres were red, green and blue with the same $\psi^{i,j}$ of $57.74^{\circ }$, while the other three spheres were yellow, cyan and magenta with $\psi^{i,j}$ of $35.26^{\circ }$. The 32 light positions were configured with r=442 mm and $\theta =20^{\circ }$. The reference plane was located where $R_{W}=diag\left(\left[1,-1,-1\right]\right)$ and $t_{W}={\left[0,0,678\right]}^{T}$. $k_{d}^{i,j}$, $k_{s}^{i,j}$ and $\beta^{i,j}$ were set the same across the field of view as 0.4, 0.2 and 100, respectively. Additive Gaussian noise was introduced with $\sigma_{n}=0.02$. The image irradiance under the first illuminant is shown by Figure 4a. The parameters of the proposed PS method were configured as: $T_{d}=0.01$, $T_{c}=5^{\circ }$, $T_{o}=2.5$, $T_{m}=0.0036$, $T_{\alpha }=3$, $T_{x}=0.02$ and $T_{y}=0.02$. 100 repeated tests were conducted and the mean value was adopted for evaluation.

Figure 4b shows the angular error of surface orientation estimation without surface normal refinement. As is shown, the central regions of the spheres where dense non-Lambertian reflectances appear have larger error. By including the proposed surface normal refinement step, the angular errors of surface orientations around the detected dense non-Lambertian reflectance region, shown by a ring, are significantly reduced as shown by Figure 4c. This is where the reflectance parameters are most tuned through the surface normal refinement. Errors in such regions are even smaller than those with sparse non-Lambertian reflectance, which further enhances that making specularities as meaningful signals is beneficial for surface orientation estimation. Figure 4d demonstrates the improvement percentage for quantitative evaluation. The improvement percentage is defined reduction of the error divided by the error without surface normal refinement. The accuracy of surface orientations was enhanced by 34.33% in median and 32.25% on average. The first and third quantile values of the improvement were 15.76% and 54.23%, respectively. The performance improvements onto the six different-coloured spheres are shown in Table 1. The improvements were obvious for all spheres is significant, implying that the method was applicable to a wide range of surfaces with different spectral reflectances. For spheres that had larger chromatic angle, $\psi^{i,j}$, the ${\bar{d}}_{RGB}^{i,j}$ estimations were more accurate. The better estimated ${\bar{d}}_{RGB}^{i,j}$ led to the more accurate recovery of ${\left({\underline{f}}_{s}^{i,j}\right)}_{1}$, resulting in the more reliable initialisation of ${\left({\bar{n}}^{i,j}\right)}_{1}$. Such initialisations of ${\bar{n}}^{i,j}$ ultimately affected the different improvement performances. Overall, the accuracy was enhancement by including the additional surface normal refinement step and for spectral reflectances with larger chromatic angles, the improvement was more phenomenal.

In the second experiment, the performance improvement due to surface normal refinement was verified onto a wide range of reflectances. Experimental settings and parameters of the proposed method were the same as the first experiment except that different values of $k_{d}^{i,j}$, $k_{s}^{i,j}$ and $\beta^{i,j}$ were applied to represent various surface reflectances when generating synthetic images. The ratio of $\kappa_{ds}^{i,j}=k_{d}^{i,j}/k_{s}^{i,j}$ represents the relative strength between the diffuse and specular components, whereas $\beta^{i,j}$ indicates the width of the specular lobe. The mean, median, first and third quantile values of the improvement were adopted for evaluation and the angular error of ${\bar{d}}_{RGB}^{i,j}$ estimation was used for analyses. As shown from Table 2, the improvements on all the different reflectances were significant and more obvious for larger value of $\beta^{i,j}$ and $\kappa_{ds}^{i,j}$. Larger value of $\beta^{i,j}$ suggesting narrower specular lobe made the ${\bar{d}}_{RGB}^{i,j}$ estimation more accurate since the sparse non-Lambertian reflectance assumption was valid. Similarly, larger value of $\kappa_{ds}^{i,j}$ indicating stronger diffuse components made better ${\bar{d}}_{RGB}^{i,j}$ estimation due to stronger inliers. More accurate ${\bar{d}}_{RGB}^{i,j}$ estimation led to bigger improvement from the surface normal refinement. In summary, with the additional surface normal refinement step, the accuracy of surface orientation estimation is enhanced by around 30% on average and the improvement is more obvious for reflectance that has narrower specular lobe and stronger diffuse component.

The third experiment aims to evaluate the overall performance of the proposed colour PS method onto a wide variety of dielectric material reflectances. Twenty-four material BRDFs divided into six categories were evaluated, where the different BRDFs were provided from the MERL database [33]. Experimental settings and the method parameters were exactly the same as the first experiment except that the BRDFs were replaced. The image irradiance for the first six materials under the first illuminant is shown in Figure 5a. The angular errors of the surface orientation estimation are given in Figure 5b. As is shown, the error is not only larger at the near central regions where specularities overlap but also at the boundaries. Boundary pixels in the spheres have shallow grazing angles and the error is due to the lack of modeling of the fresnel reflection in the Blinn-Phong model. Figure 5c shows the performance of surface normal estimation on the 24 materials using box-and-whisker plot. The red and black dot represent the mean and median value, respectively. The lower and upper bound of the box indicates the first and third quantile values. The six colours of the boxes suggest the six material categories. As is shown, the proposed method can accurately estimate ${\bar{n}}^{i,j}$ for most dielectric materials within 5 degrees on average. The estimation performances on phenolic, plastic and rubber were consistent, whereas those on the wood stain, fabric and acrylic were also acceptable with two exceptions, the violet acrylic and the green fabric. The degradation for violet acrylic was due to its wide specular lobe with small $\kappa_{ds}^{i,j}$, while the reason for green fabric was because of its small $\psi^{i,j}$.

### 4.2. Evaluations Using Real Images

The proposed method was evaluated on five different datasets comprised of real images in the DiLigenT database [4]. Parameters of the proposed method were set the same as the first experiment. ${\bar{s}}_{RGB}$ was estimated as ${\left[0.5774,0.5774,0.5774\right]}^{T}$. From the first to fifth column in Figure 6, the results on BUDDHA, BEAR, POT2, READING and GOBLET are respectively shown. Figure 6a shows the image irradiance under the first illuminant, while Figure 6b,c demonstrate the estimated normal map and the angular error of surface orientation estimation, respectively. As is shown from READING, the proposed method is only feasible for surfaces whose ${\bar{d}}_{RGB}^{i,j}$ and ${\bar{s}}_{RGB}$ are distinct. Figure 6c shows that the dense non-Lambertian problem where the specularity overlaps is mostly solved for BUDDHA, BEAR and POT2, while deficits still exist on GOBLET, which is because the DRM is not proper to characterize metallic reflectance.

The results were compared with those of nine other PS methods which are listed in Table 3. The results from the nine methods were provided from the DiLigenT database and these methods were representative and well-recognized. It is to be noted that the results of the recent machine learning based techniques reviewed in Introduction are not shown for comparison since the effect of the physics based models is investigated in this paper. Due to the data availability, refs. [21,23] were not involved in the comparison, while the essence of their works in using DRM with known ${\bar{s}}_{RGB}$ was well-inherited by the proposed method.

Figure 6d shows the results from ten PS methods on the five datasets. The X-axis represents the method index and the Y-axis indicates the angular error of ${\bar{n}}^{i,j}$. The five colours of the bar represent the different datasets. The angular error of ${\bar{n}}^{i,j}$ is also demonstrated using the box-and-whisker plot similar to Figure 5. As is shown, the proposed method performs well in estimating surface orientations for all the five datasets except GOBLET. The results suggest that modeling reflectance using DRM is proper for surfaces made of dielectric materials. The strength of the proposed method lies on the reliable estimation of ${\bar{d}}_{RGB}^{i,j}$ with simultaneous specularity detection. The advantage is also accredited by exploiting ${\bar{n}}^{i,j}$ not only from the diffuse components but also from the specularities owing to the diffuse-specular separation. In spite of its advantages, the proposed method is not effective for metallic surfaces compared with methods, such as [14]. The limitation originates from the fundamental assumption of DRM that ${\bar{s}}_{RGB}$ are the same across the surface. This assumption is violated for metallic surfaces since the wavelength and geometry exhibit inter-dependency.

## 5. Conclusions and Future Works

A PS method using colour images dealing with non-Lambertian reflectance has been proposed. The method formulates the imaging photometry using DRM with known specular colour. It extracts surface orientations not only from the diffuse components but also specularities owing to the diffuse-specular separation. Introducing the additional surface refinement step using information from specularities, the proposed method particularly improves the accuracy for surface orientation estimation at pixels where dense non-Lambertian reflectance appear. The simultaneously acquired specular parameters can be applied for more potential functionalities, such as digital relighting and material classification.

From the experiment of validating the effectiveness of the newly proposed surface normal refinement step on surface where dense non-Lambertian reflectance appears, the results indicate that with the additional step, the accuracy is enhanced by around 30% on average and the improvement is more phenominal for surfaces with larger chromatic angles, stronger diffuse components and narrower specular lobes due to the better estimated unit diffuse colour. The result investigating the proposed method applicability on 24 different reflectances of dielectric materials suggests that the average angular error of surface orientation estimation for most materials are within $5^{\circ }$ and bigger errors occur at surface patches with shallower grazing angles due to the limitation of the Blinn-Phong model that does not incorporate the fresnel reflection. From the systematic comparison on five datasets with nine other representative methods, the proposed method shows its descent performance on reflectances of dielectric materials and degradation on metallic surface due to the limitation of DRM. The result has also demonstrated that the proposed method is only feasible for surfaces whose diffuse and specular colour are distinct.

This paper presented the first set of results to estimate surface orientations using colour images for non-Lambertian reflectance, while it can be extended in a variety of ways. First, other analytical BRDFs can be employed instead of the Blinn-Phong model to better account for the fresnel reflection. Second, machine learning based models can be incorporated to learn and reduce errors of analytical models. Further, an alternative method can be proposed specifically for surfaces whose diffuse and specular colours are close to extend the method’s applicability. Making the method adaptive is one possible solution. Last but not least, it is also of particular interest to infer surface properties from the estimated specular parameters for the purpose of material classification.

## Figures and Tables

**Figure 1 jimaging-08-00040-f001:**
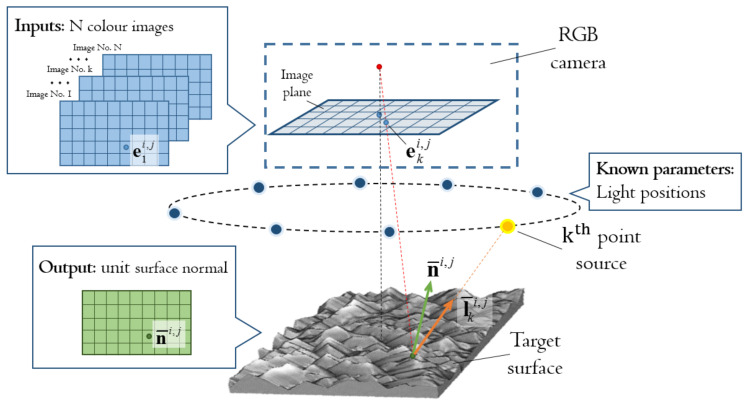
Schematic diagram for generic colour PS.

**Figure 2 jimaging-08-00040-f002:**
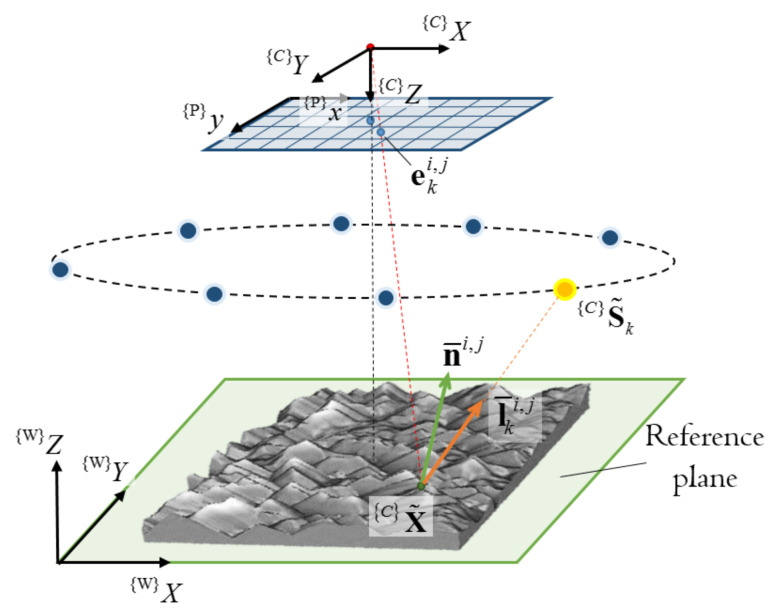
Coordinate setups for modeling imaging geometry and light configuration.

**Figure 3 jimaging-08-00040-f003:**
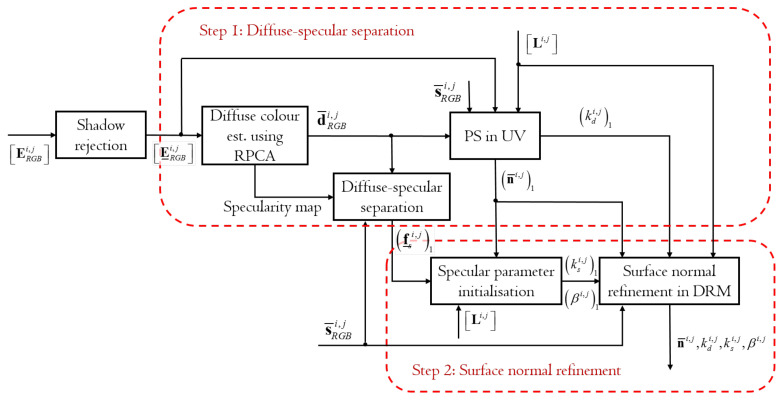
Flows and the original contribution of the DRM-based colour PS.

**Figure 4 jimaging-08-00040-f004:**
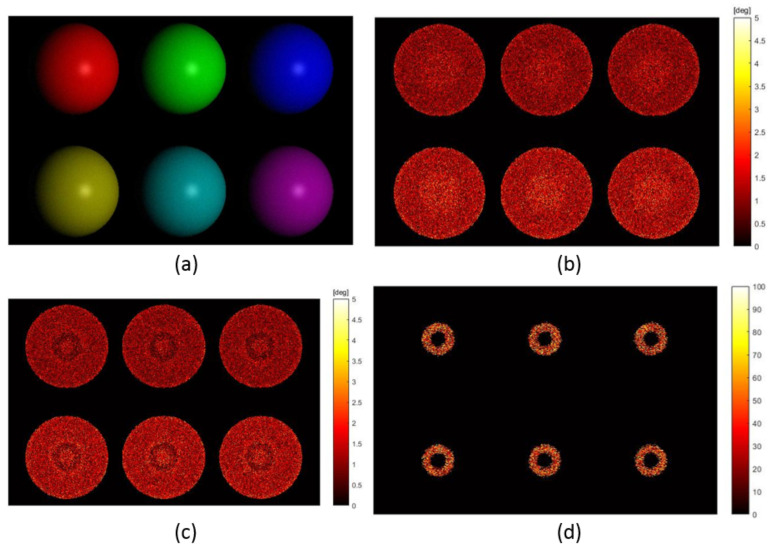
Evaluation of surface normal refinement: (**a**) Image irradiance under the first illuminant; (**b**) Angular error of surface orientations without surface normal refinement in degrees; (**c**) Angular error of surface orientations with surface normal refinement in degrees; (**d**) Improvement of surface orientation estimation by including surface normal refinement in percentage.

**Figure 5 jimaging-08-00040-f005:**
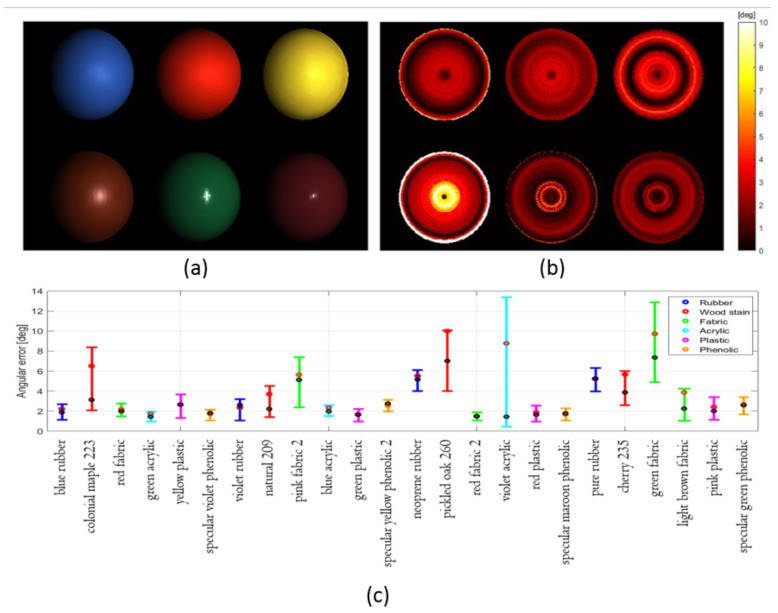
(**a**) Image irradiance for the first six material BRDFs under the first illuminant; (**b**) Angular error of surface orientation estimation for the first six material BRDFs; (**c**) Angular error of surface orientation for the twenty-four dielectric materals.

**Figure 6 jimaging-08-00040-f006:**
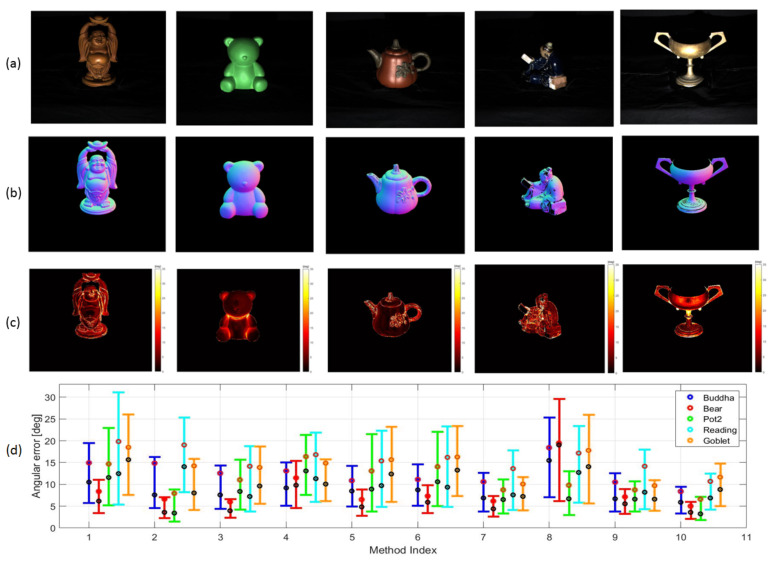
Evaluations on surface oreintation estimation: (**a**) Image irradiance under the first illuminant; (**b**) Estimated normal map; (**c**) Angular error of surface orientations; (**d**) Angular error of surface orientations on five datasets using ten different PS methods.

**Table 1 jimaging-08-00040-t001:** Improvement on different-coloured spheres.

Index	Colour	$\psi^{i,j}$	Error of ${\bar{d}}_{{\mathrm{RGB}}^{i,j}}$	Mean Improvement
1	red	$57.74^{\circ }$	$1.15^{\circ }$	23.39%
2	yellow	$35.26^{\circ }$	$1.33^{\circ }$	32.85%
3	green	$57.74^{\circ }$	$1.15^{\circ }$	23.42%
4	cyan	$35.26^{\circ }$	$1.32^{\circ }$	32.04%
5	blue	$57.74^{\circ }$	$1.15^{\circ }$	23.15%
6	magenta	$35.26^{\circ }$	$1.33^{\circ }$	32.24%

**Table 2 jimaging-08-00040-t002:** Evaluation of the improvement for different reflectances.

$k_{d}/k_{s}$	β	Mean	Median	First Quantile	Third Quantile	Error of ${\bar{d}}_{\mathrm{RGB}}$
0.4/0.2	100	32.25%	34.33%	15.76%	54.23%	$1.23^{\circ }$
0.4/0.4	100	31.14%	32.97%	15.21%	53.75%	$1.58^{\circ }$
0.4/0.8	100	30.48%	32.77%	14.27%	52.90%	$2.34^{\circ }$
0.4/0.2	20	27.49%	26.70%	6.80%	51.66%	$2.99^{\circ }$
0.4/0.4	20	25.92%	25.98%	6.67%	51.25%	$5.06^{\circ }$
0.4/0.8	20	25.02%	24.07%	6.57%	50.81%	$8.30^{\circ }$

**Table 3 jimaging-08-00040-t003:** Comparative studies of PS methods.

Method Index	1	2	3	4	5	6	7	8	9	10
Reference	[3]	[9]	[11]	[12]	[19]	[20]	[13]	[34]	[14]	proposed

## Data Availability

Not applicable.

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
