# Peer review of "DRM-Based Colour Photometric Stereo Using Diffuse-Specular Separation for Non-Lambertian Surfaces"

_2313-433X, 2022, doi:10.3390/jimaging8020040_

Round 1

Reviewer 1 Report

In this paper, a BRDF-based photometric stereo (PS) algorithm is proposed to deal with a non-Lambertian reflection. The performance of BRDF-based PS is quite dependent on the initial value of the parameters such as diffuse coefficients, specular coefficients, surface roughness, and the surface normal where the Blinn-Phong BRDF model is adopted to represent the surface appearance of the dichromatic materials; The diffuse-specular separation using the robust principal component analysis and the dichromatic reflectance model is exploited to obtain the reliable initial values of these parameters. Then, an iterative nonlinear least-square optimization is applied to refine surface normals.

This manuscript does not reflect the recent work on the photometric stereo, and the most recent reference is a paper published in 2016. In addition, there is no comparison with state-of-the-art PS methods. Deep learning algorithms have recently achieved remarkable progress in computer vision. Following this trend, deep learning-based approaches (MLP, CNN, and GNN) in the photometric stereo have been proposed to estimate accurate surface normal. Nevertheless, there is no description of deep learning-based methods in this paper. A literature study on the deep learning-based method should be included in the introduction section. In addition, the deep learning-based methods have improved accuracy by a large margin compared to the methods mentioned in this paper. The comparison of the proposed method with the deep learning-based methods for the DiLigenT PS benchmark database used in this paper should also be included.

Author Response

Please see the attachment (rejoinder and revised manuscript).

Reviewer 2 Report

The manuscript presents a method to estimate surface orientations using diffuse specular separation dealing with non-Lambertian reflectance. The method is tested with synthetic images and images from a benchmark data set. I find the manuscript well-written and the results seem promising.

Some comments:

line 51: unclear what you mean by "initialisations"?

line 66: do you mean Lambertian or non-Lambertian effects that are treated as outliers? unclear: what exactly is treated as outliers?

Please add a list of acronyms and abbreviations to increase readability. I acknowledge that all acronyms are defined before use, and these are commonly used, but as there is a high number of these, this list would be really helpful.

line 127: strange sentence: "... or the UV colour space ...". maybe " ... i.e., the UV colour space as S is not considered". (remove "more exactly") When not considering S, is then S normalised?

Figure 3: Please enlarge the image, as the fonts are partially too tiny. The shadow-rejection, is it related to the outliers? Further, is this algorithm per light source?

Please enlarge Figure 4. Why is the improvement visible as a ring, i.e., only for a specific range of angles on the sphere?

Line 275: unclear what you mean by "The accuracy was enhanced by 34.33% ...". And which of the images in Fig 4 does this value relate to? What is the formula to calculate the percentages?

line 278: vague: "The improvements were obvious ..."

Figure 5: please enlarge both parts of this figure. Please avoid yellow in Fig. 5c, as it is only barely visible in print.

Figure 6: Please enlarge this figure. Please also note that yellow on white in Fig 6d is not very visible!

line 380: the further work sketched here is too generic. Please also mention some ideas on how one would attempt to solve these new research questions.

The conclusion section is rather generic.

Author Response

(The authors gave the same response as above.)

Round 2

Reviewer 1 Report

In this paper, a BRDF-based photometric stereo (PS) algorithm is proposed to deal with a non-Lambertian reflection.

The reviewer recommends accept after minor revision for the revised manuscript.

Author Response

Thank you for the recommendation of acceptance after minor revision.  Since no new comments or suggestions were given, we believe the reviewer meant acceptance after the "last" minor revision (though we made a reasonable amount of revision to reflect all the reviewers' comments).  Since the reviewer selected "the conclusions supported by the results can be improed", we added a revision.  For reference, I have attached the updated rejoinder and revised paper.  The revision has improved all the introduction, the results and the conclusions.  
